# CONTROLLING VIDEO GENERATION WITH VISION LANGUAGE MODELS

## ABSTRACT

Controlling video generation models typically requires finetuning on video datasets with explicit control labels. However, collecting such datasets is costly, and the control modality in the data inherently restricts the controllability of the trained models. In contrast, vision language models (VLMs) can readily generalize to new tasks with pretrained knowledge and in-context learning. Motivated by this capability, we introduce Ask-A-Video, a test-time training paradigm that formulates controllable video generation as visual question answering (VQA): a video generator produces video frames, a frozen VLM answers control-related questions, and the VQA loss is directly backpropagated to the video generator. By leveraging the generalization of VLMs, Ask-A-Video enables efficient and flexible control for any off-the-shelf video generator without the need for any video data. Empirically, our method improves controllability for both text-to-video and image-to-video models across different families and scales. Compared to adding constraints via prompt extension, Ask-A-Video yields stronger prompt following and more physically plausible dynamics. It also enables fine-grained spatial and motion control through visual prompting. In addition, since our method distills controllability into the model weights, it allows reusing the learned control for new prompts without additional cost.

## 1 INTRODUCTION

Controllable video generation has huge potential in movie-making, game creation, and embodied agents. However, allowing users to explicitly control frame content, object movement, and physical consistency remains a fundamental challenge. Most existing video generation models (Brooks et al., 2024; Kong et al., 2024; Wan et al., 2025) only provide coarse, high-level control like text prompts. To achieve fine-grained control, many approaches add additional conditioning modules to the video generator and finetune on large, carefully curated video datasets paired with explicit control labels (*e.g.*, camera trajectories, sketches, depth, and optical flow) (Cheong et al., 2024; He et al., 2024; Zeng et al., 2024; Liu et al., 2025b; Xing et al., 2024; Shi et al., 2024; Xing et al., 2025). Creating such datasets is expensive, and the type of control encoded in the data naturally restricts how the trained video generation model can be controlled. Unlike video generators, which are optimized for synthesizing realistic pixels, vision-language models (VLMs) (OpenAI, 2023; Bai et al., 2025) are equipped with strong visual reasoning and flexible prompting abilities. These properties make VLMs a natural choice for serving as powerful judges in diverse controllable video generation tasks.

However, most prior attempts to incorporate VLMs for video generation at inference time are limited to prompt engineering or non-differentiable sampling and re-ranking (Liu et al., 2025a; Xue et al., 2025). These strategies often struggle to provide fine-grained control and scale poorly with sampling budgets. Inspired by this gap, we propose Ask-A-Video, a test-time training paradigm that formulates controllable video generation as a visual question answering (VQA) problem. This formulation utilizes the visual recognition and reasoning ability of VLMs as a *differentiable* supervision signal to guide the video generators. Desired controls are expressed as questions about the generated video, such as prompt adherence (*"Does the video contain a red sports car facing left?"*), spatial layout (*"Is the dog inside the boxed region?"*), or physics plausibility (*"Is the motion consistent with gravity?"*). This formulation offers three practical advantages. First, since VQA serves as a *general control* interface, it enables free-form control without additional control modules. Any modality that a VLM can condition on (*e.g.*, text, images, bounding boxes, or multi-step instructions)

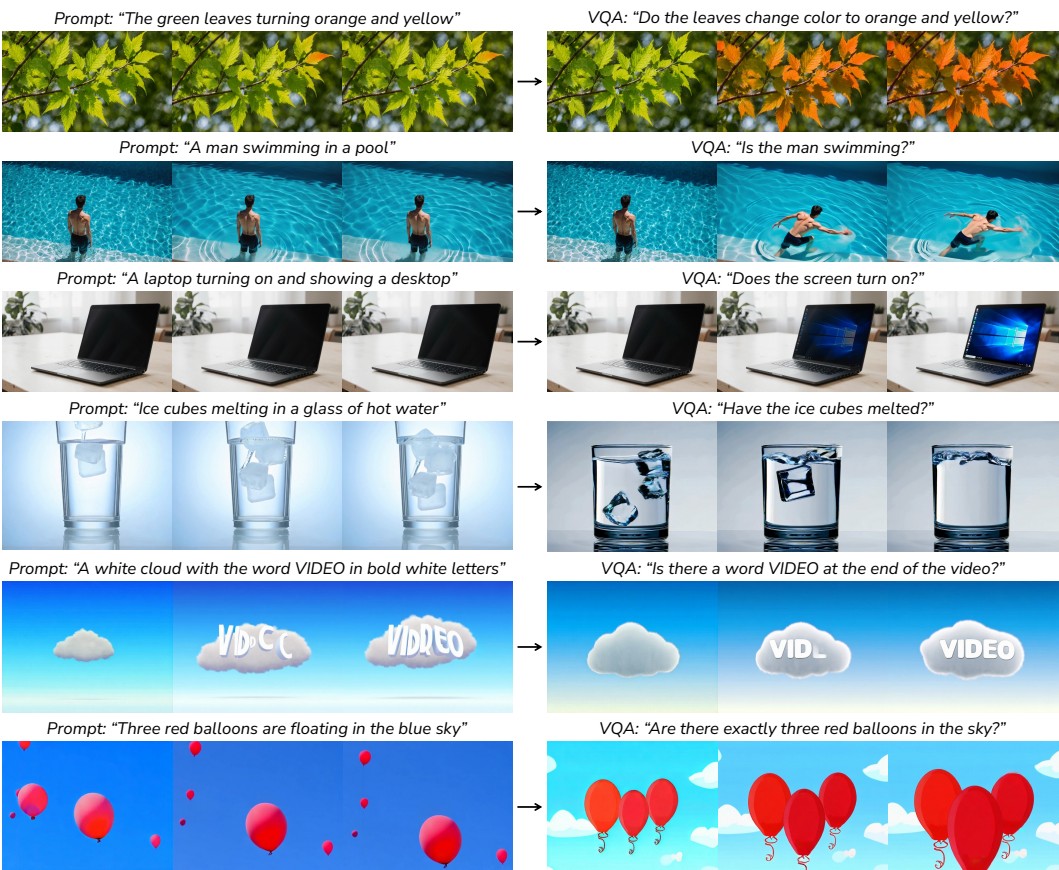

Figure 1: Ask-A-Video can improve the prompt following, physical plausibility, and text rendering of any existing video generator by leveraging pretrained VLMs, without video training data.

can be naturally expressed as a question-answer pair. Second, the training process is *video-free*: rich learning signals can be obtained by directly backpropagating the VQA loss from the VLM to the video generator, without relying on videos or control labels. Third, the approach is *model-agnostic* and can be applied post hoc to any video generator, regardless of its training and inference recipe.

To instantiate this formulation on modern video diffusion models, we develop a forward strategy aligned with multi-step denoising, an efficient pixel-to-latent backpropagation through the video VAE, and differentiable visual prompting for fine-grained control. Specifically, given a text or visual prompt and a set of question-answer pairs describing desired properties, we iteratively finetune the video generator to minimize the VQA loss. At each iteration, we first sample a random denoising timestep and forward the generator in a no-gradient mode to obtain the partially noisy latent. Then, we perform a differentiable forward at that timestep and obtain the clean video estimate. This ensures training-inference consistency and reliable VLM supervision even at high-noise timesteps. The latent is decoded into frames through the video VAE and fed into the VLM. The likelihood of predicting the correct answer provides a differentiable supervision signal that is backpropagated through both the VLM and the video VAE into the generator, with memory efficiency ensured by slicing, tiling, and gradient checkpointing in the VAE.

We validate the effectiveness of our approach through extensive experiments on both text-to-video (T2V) and image-to-video (I2V) settings across different video generator families and sizes (Figure 1). Compared to adding control via prompt extension, we demonstrate consistent gains in controllability for both T2V and I2V, including prompt adherence in human activities and object dynamics, as well as physics plausibility for diverse scenarios (*e.g.*, gravity, collision, and reflection). Our method also supports fine-grained spatial and motion control through visual prompting. Moreover, it distills controllability into the generator's weights, enabling the learned behavior to generalize to new prompts without additional optimization.

## 2 BACKGROUND

### 2.1 VIDEO DIFFUSION MODELS

Modern video generation models (Brooks et al., 2024; Gao et al., 2025; Kong et al., 2024; Wan et al., 2025) are commonly formulated as diffusion processes that progressively transform Gaussian noise into the target data distribution (Sohl-Dickstein et al., 2015; Ho et al., 2020; Song et al., 2020). Flow matching provides a conceptually simpler yet equivalent formulation and has become a popular paradigm (Chen et al., 2018; Albergo & Vanden-Eijnden, 2022; Liu et al., 2022; Lipman et al., 2022; Esser et al., 2024). During training, given a video $x_0$ from the training data, a timestep $t \in [0, 1]$ is sampled from a logit-normal distribution, and noise is initialized as $\epsilon \sim \mathcal{N}(0, I)$. The training sample $x_t$ is constructed by linear interpolation

$$x_t = (1 - t)x_0 + t\epsilon.$$

A video generation model parameterized by $\phi$ is commonly trained to predict the velocity $u = dx_t/dt = \epsilon - x_0$ by minimizing the mean squared error between the estimated velocity $\hat{u}$ and $u$:

$$\mathcal{L}(\phi) = \mathbb{E}_{t,x_0,\epsilon}\left\|\hat{u} - u\right\|_2^2.$$

During inference, a noise sample $\epsilon \sim \mathcal{N}(0, I)$ is drawn, and a solver integrates the estimated field backward in time $x_{t'} = x_t + (t' - t)\hat{u}$ for $t' < t$ to obtain the sample $x_0$ at $t = 0$.

To make training and sampling tractable at high resolution and frame rate, most video diffusion models (Brooks et al., 2024; Kong et al., 2024; Wan et al., 2025) use a spatio-temporal causal VAE to compress raw pixels $x \in \mathbb{R}^{T \times H \times W \times 3}$ into a compact latent space $z \in \mathbb{R}^{(\frac{T-1}{c_t}+1) \times \frac{H}{c_s} \times \frac{W}{c_s} \times C}$, where $c_t$ and $c_s$ are temporal and spatial compression ratio, respectively. This compression significantly reduces the number of tokens consumed by diffusion transformers.

### 2.2 VISION LANGUAGE MODELS

VLMs (Liu et al., 2023; OpenAI, 2023; Gemini Team, 2023; Wang et al., 2023; Bai et al., 2025) jointly model visual and text modalities within a unified autoregressive framework. They demonstrate strong generalization on tasks such as visual question answering and video understanding. Given a visual condition $v$ such as an image or a video and a text token sequence $y = (y_1, \ldots, y_T)$, the training follows the same next-token prediction objective as standard language models:

$$\mathcal{L}(\theta) = -\sum_{t=1}^{T} \log p_\theta(y_t \mid y_{<t}, v),$$

where $\theta$ denotes the VLM parameters, $T$ is the length of the output sequence, $y_t$ is the $t$-th target token, and $y_{<t}$ is the prefix of previously generated tokens. By concatenating exemplar inputs and outputs or an explicit chain of thought into the prompt, VLMs can quickly acquire new task formats without parameter updates and thereby adapt to diverse objectives. Moreover, users may annotate the visual inputs with boxes, points, or arrows to indicate regions of interest, providing richer signals than text-only instructions and improving visual perception and reasoning.

## 3 METHOD

Our approach formulates controllable video generation as visual question answering by leveraging the generalizability and flexibility of VLMs (Sec. 3.1). Given a text prompt or visual prompt, and question-answer pairs, our approach iteratively finetune the video generator to generate video frames that minimize the VLM loss for prediction of the correct answer of the question. At each iteration, we sample a random denoising timestep to forward the video generator, and estimate the clean video frames at that timestep as VLM input (Sec. 3.2). Because the VLM loss is pixel-level while the video generator operates in a latent space, we design a memory-efficient strategy to backpropagate loss from the pixel space to the latent space through the video VAE (Sec. 3.3).

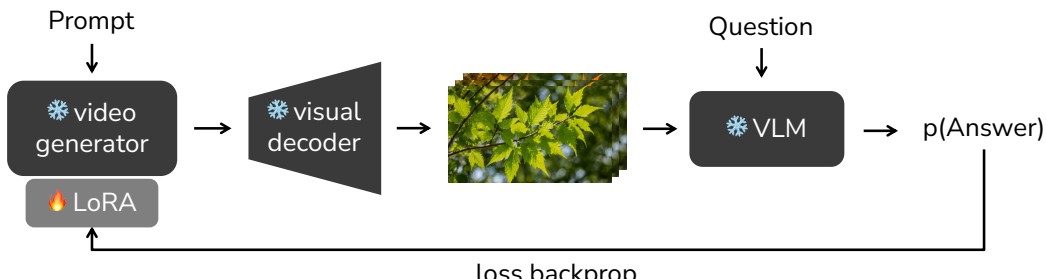

Figure 2: We frame controllable video generation as a visual question-answering task. Starting with latents produced by the video generator, we decode them into frames using a video VAE and feed them into the VLM with questions. The VLM's loss from predicting the correct answer is backpropagated through the VLM and the VAE to the generator. As a result, we directly finetune the video generator to produce videos that maximize the likelihood of yielding the target answer.

## 3.1 CONTROLLABLE VIDEO GENERATION AS VISUAL QUESTION-ANSWERING

Let a generated video in pixel space be $x_0 \in \mathbb{R}^{T \times H \times W \times 3}$ and its latent representation be $z \in \mathbb{R}^{(\frac{T-1}{c_t}+1) \times \frac{H}{c_s} \times \frac{W}{c_s} \times C}$ as in Sec. 2.1. We index frames by $n \in \{1, \ldots, T\}$. The generator with parameters $\phi$ produces a latent trajectory $\{z_t\}_{t \in [0,1]}$ and we decode the estimate at $t = 0$ to pixels via the frozen decoder $D_\psi$:

$$\hat{x}_0 = D_\psi(\hat{z}_0).$$

Given a question $q$ and target answer token sequence $y = (y_1, \ldots, y_L)$, the VLM with parameters $\theta$ defines the next-token likelihood $p_\theta(\cdot)$. The loss for a single QA pair on the video is

$$\mathcal{L}_{\text{VQA}}(\phi) = -\sum_{\ell=1}^{L} \log p_\theta(y_\ell \mid y_{<\ell}, \hat{x}_0, q),$$

where gradients flow into $\phi$ through $D_\psi$ while $\theta$ and $\psi$ remain frozen.

Beyond textual control, our method also allows fine-grained control through visual prompting. For any subset $\mathcal{S} \subseteq \{1, \ldots, T\}$, frame $n \in \mathcal{S}$ is paired with a prompt image $p_n$ (*e.g.*, keypoints, boxes, masks, or arrows) and a soft mask $\alpha_n \in [0, 1]^{H \times W}$. We form a differentiable alpha blend

$$\tilde{x}_0^{(n)} = (1 - \alpha_n) \odot \hat{x}_0^{(n)} + \alpha_n \odot p_n, \qquad n \in \mathcal{S},$$

and replace $\hat{x}_0^{\mathcal{S}}$ with $\tilde{x}_0^{\mathcal{S}}$ in the objectives above, where $\odot$ denotes element-wise multiplication. This provides intuitive spatial/motion control while keeping gradients consistent.

## 3.2 FORWARD PROCESS

We follow the flow-matching parameterization in Sec. 2.1, where $x_t = (1-t)x_0 + t\epsilon$ and the velocity is $u(t) = \epsilon - x_0$. In latent space we use the same convention, letting the video generator predict $\hat{u}_\phi(z_t, t)$. During each training iteration, we sample a solver time grid $\{t_i\}_{i=0}^{K}$ with $1 = t_0 > t_1 > \cdots > t_K \approx 0^+$ and draw $\epsilon \sim \mathcal{N}(0, I)$. In stop-gradient mode, we run the same noise scheduler and ODE solver used at inference to ensure train-test consistency:

$$\tilde{z}_0^{(k)} = \text{Solver}(\hat{u}_\phi; z_{t_0} = \epsilon, t_0 \to t_k), \quad k \sim \text{Uniform}\{0, \ldots, K-1\}.$$

We then perform a single differentiable forward pass at the same $t_k$ to obtain gradients with low memory. Specifically, we re-noise

$$\tilde{z}_{t_k} = (1 - t_k)\tilde{z}_0^{(k)} + t_k\epsilon, \qquad \epsilon \sim \mathcal{N}(0, I),$$

evaluate the velocity (for $v$-parameterization)

$$u(t) = \epsilon - z_0, \qquad \hat{u} = \hat{u}_\phi(\tilde{z}_{t_k}, t_k),$$

and form the cleaned latent estimate

$$\hat{z}_0 \;=\; \tilde{z}_{t_k} - t_k\,\hat{u}.$$

Finally, we decode $\hat{z}_0$ to pixels via the frozen decoder $D_\psi$ and compute the VLM objective on $\hat{x}_0 = D_\psi(\hat{z}_0)$. We use UniPC (Zhao et al., 2023) as the high-order predictor-corrector solver to reduce bias and stabilize supervision at high noise levels, but the scheme is compatible with diverse schedulers used in modern video generation models.

### 3.3 Backpropagation from Pixel to Latent Space

Directly backpropagating VQA losses through the video VAE decoder easily exceeds the memory limits when handling long video sequences with high spatial resolutions. To address this challenge, we design a memory-efficient strategy based on frame-wise decoding, spatiotemporal tiling, slicing, and gradient checkpointing, ensuring that peak memory does not scale with the total video length $T$. Under this design, pixel-space gradients can still be faithfully propagated back to the corresponding latent slices. Formally, given a latent tensor $z$, we first apply a temporal sliding window of length $K_t$ with overlap $O_t$ to obtain subsequences $\{z^{(i)}\}$. Within each subsequence, we further perform spatial tiling with window size $K_h \times K_w$ and overlaps $(O_h, O_w)$. Each tile is decoded by the frozen VAE decoder $D_\psi$ with gradient checkpointing and CPU offloading enabled to produce partial reconstructions $\{\hat{x}^{(i,j)}\}$. Overlapping regions are blended using linear ramp weights. Along the temporal axis, for overlap length $\Delta_t$ we apply weights $w_t(x) = x/\Delta_t$. Along the spatial axes, we construct one-dimensional ramps along height and width with overlaps $\Delta_h, \Delta_w$, and take the elementwise minimum as the per-pixel weight. These operations ensure smooth transitions and spatiotemporal continuity. Importantly, all slicing, tiling, and blending procedures remain fully differentiable, allowing pixel-level gradients to backpropagate to the latent space.

For the VLM input, we further subsample frames by selecting a fixed index set $\mathcal{T} = \{n_1, \ldots, n_K\} \subseteq \{1, \ldots, T\}$, and rescale them to the required input resolution using differentiable interpolation. The supervision signals from these frames propagate back through the interpolation and framewise decoding steps, and update the generator parameters $\phi$. This strategy bounds peak memory by the chosen window/tile sizes rather than the full spatiotemporal extent $(T, H, W)$, while retaining end-to-end gradient flow and temporal coherence.

## 4 Results

We demonstrate that many controllable video generation tasks can be accomplished within our simple framework, which can improve prompt adherence and physical plausibility. We also demonstrate advanced fine-grained control via visual prompting. In addition, we show that a previously trained checkpoint can directly generalize to other inputs without additional training.

### 4.1 Experimental Details

We apply our framework to two representative families of diffusion-based video generators: Wan2.1-T2V-1.3B (Wan et al., 2025) for text-to-video generation, producing 81 frames at $640 \times 480$ resolution; FramePack-F1 (Zhang & Agrawala, 2025), a variant of HunyuanVideo-13B (Kong et al., 2024), for image-to-video generation, producing 25 frames at $832 \times 480$ resolution. We evaluate on both text-to-video (T2V) and image-to-video (I2V) generation with curated 50 in-house prompts covering diverse domains and sampled 3 random seeds per task. Prompts are grouped into three categories: human activities (*e.g.*, facial expressions, body motions), object dynamics (*e.g.*, object color, motion trajectories, appearance changes), and physics consistency (*e.g.*, collisions, gravity, thermodynamics, reflection). We use Qwen2.5-VL-7B-Instruct (Bai et al., 2025) as the frozen judge. At each iteration, we uniformly sample 4 frames from the generated video and pair them with the question-answer prompt. The VLM is queried with greedy decoding and restricted to binary answers (Yes/No) via prompt templates. We take the logit of the first answer token as the supervision signal. We finetune LoRA parameters inserted into all self-attention and cross-attention modules $(q, k, v, o)$ as well as the feedforward layers of the generator. LoRA rank is fixed across layers. For optimization, we use Adam optimizer (Adam & Ba, 2014) with learning rate $5 \times 10^{-5}$ and gradient clipping. We adopt the UniPC solver (Zhao et al., 2023) with 25 steps. All experiments are conducted on

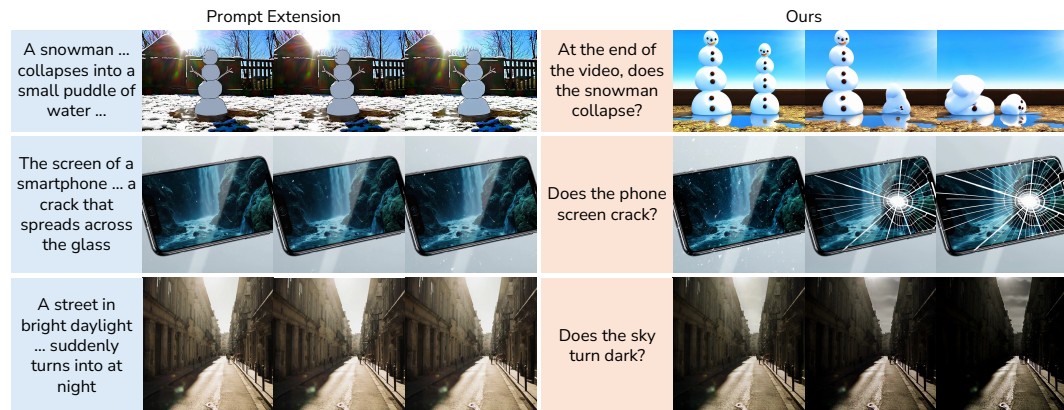

Figure 3: Compared to prompt extension, our method can further improve prompt adherence of existing video generators. This shows that VLM's knowledge is especially helpful for out-of-distribution cases of video generators.

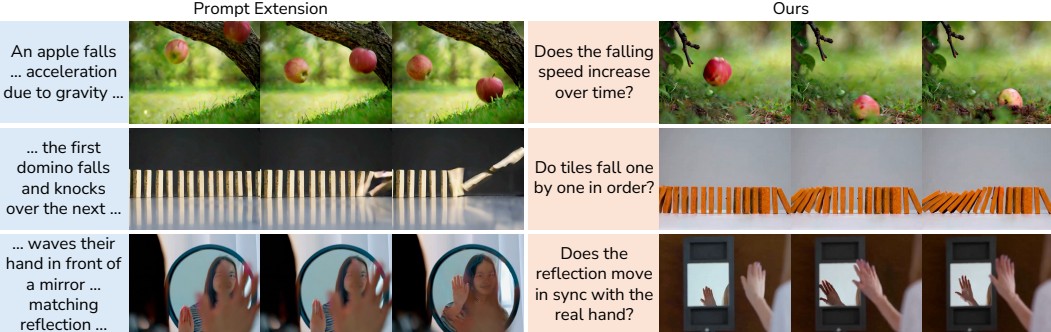

Figure 4: For scenarios that require following physical laws, our method can explicitly distill VLM's verification into the video generator to improve physical plausibility, while prompt extension does not offer noticeable enhancement.

a single NVIDIA H100 GPU, and the fastest runs finish in minutes. We use human evaluation as the primary metric. For each generated video, annotators check whether the video content satisfies the prompt and, when applicable, whether physical laws are respected. Each case is scored as 1 (satisfied) or 0 (not satisfied). We report the percentage of satisfied cases per setting. Results are broken down into human activities, object dynamics, and physical laws.

## 4.2 MAIN RESULTS

**Improving Prompt Adherence.** Our method consistently improves prompt adherence compared to both the base generators and prompt extension. As illustrated in Figure 3, our framework can better control object dynamics, human motion, and scene composition. A notable benefit is enhanced text rendering in generated videos (Figure 1), which we attribute to the VLM's ability to recognize fine-grained textual details and provide corresponding gradients to the generator. These improvements are especially clear in out-of-distribution prompts where baseline models often fail.

**Improving Physics Plausibility.** By leveraging the VLM's pretrained knowledge, our method can explicitly enforce physical constraints during generation. Figure 4 shows representative examples where gravity (falling objects), thermodynamics (melting), reflection (mirrors), and motion transfer (domino collisions) are respected more faithfully than with prompt extension. This demonstrates that the VLM judge provides a semantically meaningful signal beyond surface-level prompt matching, enabling generated videos to align better with commonsense physics.

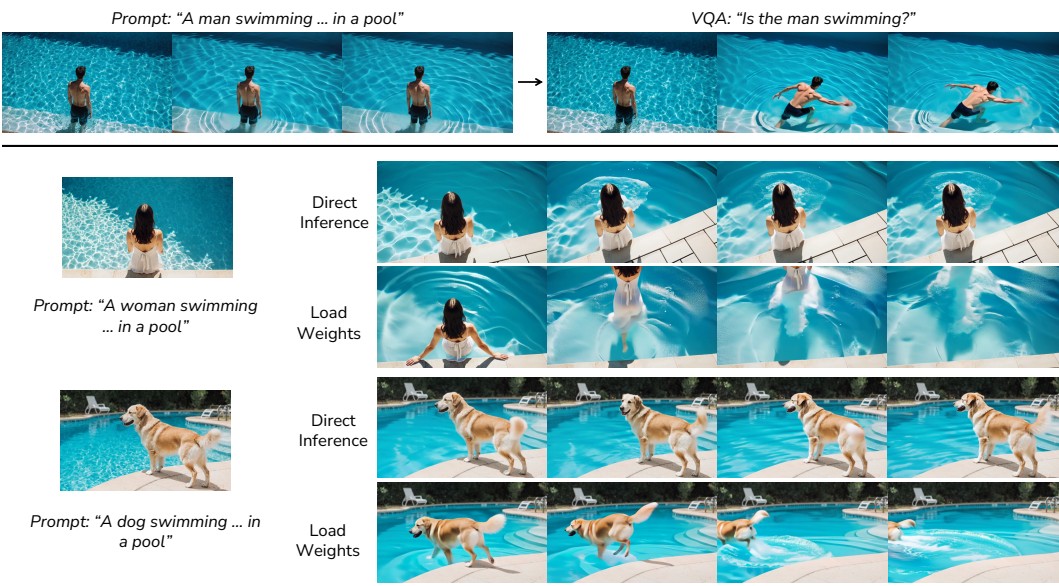

Figure 5: We leverage the flexible in-context learning ability of VLMs for visual prompting, enabling fine-grained control without adding extra modules. For example, bounding boxes can constrain action locations, and points can guide object positions.

Figure 6: Our method can generalize a previously learned controllability to other prompts without additional training by simply loading the LoRA weights, which vanilla test-time scaling cannot.

**Fine-Grained Control via Visual Prompting.** A unique advantage of our approach is its support for visual prompting. As shown in Figure 5, users can impose spatial or motion constraints through simple cues such as bounding boxes or dots. Without requiring specialized modules, the VLM seamlessly interprets these signals and translates them into effective constraints on the generator. This flexible multimodal conditioning highlights the generality of casting control as a VQA problem.

**Weight Generalization.** Unlike vanilla test-time scaling, which depends on repeated sampling, our method distills controllability into the model weights. Once learned, the LoRA weights can be reused across new prompts requiring similar controllability, eliminating the need for retraining or resampling. Figure 6 demonstrates this generalization, where a checkpoint trained for one type of control (*e.g.*, a man swimming) directly transfers to unseen prompts.

**Quantitative results.** Table 1 summarizes controllability improvements across both image-to-video (I2V) and text-to-video (T2V) tasks. Our approach achieves substantial gains on all three axes (human activities, object dynamics, and physical laws) over prompt extension. These results validate that integrating VLM feedback as a differentiable training signal is more effective than simply extending prompts.

## 4.3 FAILURE CASES

Although our approach improves controllability across diverse scenarios, we observe failure cases in situations where the generator's prior strongly contradicts the requested control or when the supervi-

Table 1: Quantitative evaluation on image-to-video (I2V) and text-to-video (T2V) tasks (%).

| Method | Human Activities | Object Dynamics | Physical Laws | Average |
|---|---|---|---|---|
| Base Prompt (I2V) | 33.3 | 9.1 | 23.1 | 20.0 |
| Prompt Extension (I2V) | 60.0 | 18.2 | 23.1 | 32.0 |
| Ours (I2V) | **86.7** | **52.2** | **53.8** | **62.0** |
| Base Prompt (T2V) | 28.6 | 31.8 | 4.3 | 18.0 |
| Prompt Extension (T2V) | **71.4** | 40.9 | 13.0 | 30.0 |
| Ours (T2V) | **71.4** | **63.6** | **52.2** | **60.0** |

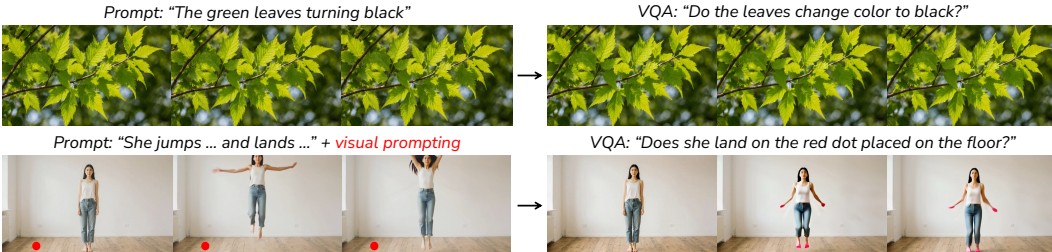

Figure 7: Compared to prompt extension, our method can further improve prompt adherence of existing video generators. This shows that VLM's knowledge is especially helpful for out-of-distribution cases of video generators.

sion signal from the VLM becomes hackable (Figure 7). For example, when asked to generate black leaves, the strong visual prior of the generator makes it difficult to explore meaningful variations, leading to suboptimal outputs. In another case, instead of following a visual prompt that required a person to land on a marked dot, the generator incorrectly colored the subject's feet red, effectively tricking the VLM into predicting the answer as correct. These examples highlight the limitations of relying solely on VLM-based supervision, especially under rare or adversarial scenarios. At the same time, they suggest that such adversarial failures could be leveraged to collect challenging examples that may further improve both the robustness of the VLM judge and the controllability of the generator. Future extensions may mitigate these issues by using stronger VLM as judges or ensembling multiple VLM judges.

## 5 RELATED WORK

**Controllable Video Generation.** To add controllability to video generation models, many recent works integrate additional conditioning signals (*e.g.*, camera movements (Cheong et al., 2024; He et al., 2024), keyboard actions (Che et al., 2025), sketches (Zeng et al., 2024; Liu et al., 2025b), depth maps (Xing et al., 2024), or motion trajectories (Shi et al., 2024; Xing et al., 2025)) into the generation process. Another line of work combines multi-modal conditions for finer control (Wu et al., 2025a; Ju et al., 2025). These diverse strategies all aim to enhance user controllability, allowing the generation of videos that more faithfully follow desired layouts, movements, or other specific constraints. Several methods leverage large language models (LLMs) to expand prompts or plan multi-frame descriptions (Huang et al., 2024; Li et al., 2024), and some use LLMs to predict entity layouts and movements (Lv et al., 2023; Lian et al., 2023), enabling layout-controllable video generation. In contrast, our approach formulates controllable video generation as a VQA problem. Any condition that can be perceived by a VLM is expressed in the form of a "question-answer" pair, without the need for extra control modules or specially annotated datasets. This enables a general control interface and a video-free training paradigm.

**Finetuning Video Generation Models.** Finetuning a pretrained video generation model on a target domain can improve the models' specific capabilities, such as controllability and long-context generation, while preserving the original model's world knowledge. This approach avoids retraining the entire network for each new scenario, enabling modular, rapid adaptation. AnimateDiff (Guo et al., 2023) introduced MotionLoRA to quickly adapt the motion module to new movement patterns. Another active direction is subject-driven and personalized video generation. Here, the goal is to

maintain a specific subject's identity or style across a generated video. Several works learn to inject a custom subject into a video model by finetuning on a few images of that subject (Wei et al., 2024; Jiang et al., 2024; Wu et al., 2025b), decoupling the learning of the subject's appearance from the learning of motion dynamics. More recent frameworks tackle multi-subject video generation (Chen et al., 2023a; Wang et al., 2024; Chen et al., 2024b), where multiple novel characters or objects must appear consistently. However, these methods often rely on large-scale, carefully curated video-control paired datasets. In contrast, our test-time training directly leverages the VQA likelihood from the VLM as a supervision signal, without requiring videos or control labels. Inference-time training has also emerged as a way to further improve generation quality on the fly, allowing the model to adjust to a given prompt or condition during the generation process. For example, Bi et al. (2025) perform optimization at inference to combine multiple concept weights (such as separate LoRA tunings for appearance and motion) into one coherent output. In comparison, we distill the controllability learned from the knowledge of VLMs into the generator's weights, enabling generalization to unseen prompts.

**VLM-as-a-Judge.** The idea originates from the "LLM-as-a-judge" approach in language modeling (Zheng et al., 2023), and is now being expanded to the multimodal domain (Chen et al., 2024a). In this paradigm, a pretrained VLM assesses the output of a visual generator against the desired conditions, and its feedback is used to improve the generator. Early implementations of this idea can be seen in works that integrate learned reward models into the training loop. For instance, Control-A-Video (Chen et al., 2023b) employs reward feedback where multiple evaluators judge the video's quality and motion consistency; the diffusion model is then optimized to maximize these rewards, leading to more faithful and smooth results. Our distinction lies in further internalizing VLM-as-a-judge into a differentiable VQA loss, which is backpropagated end-to-end into the video generator. This avoids non-differentiable re-ranking and external reward modeling. At the same time, differentiable "visual prompting" (*e.g.*, bounding boxes or keypoints) supports fine-grained spatial and motion control. Our method is aligned with the multi-step denoising process in diffusion models, ensuring training-inference consistency and efficiency. Recently, Luo et al. (2025) has explored a similar idea by using VLMs as differentiable judges for image generation. In contrast, our method operates in the video domain, which is a more challenging setting that requires modeling temporal dynamics. We address unique challenges of controllable video generation, such as video VAEs and video-specific visual prompting.

# 6 DISCUSSION

We propose a VLM-guided paradigm for finetuning video generation models, which reformulates controllable video generation as a visual question answering task. This provides a unified control interface, supports multimodal conditioning, and enables efficient test-time training without requiring video datasets. The paradigm is broadly applicable and could be extended to other types of generative models, thereby reducing reliance on costly annotated data.

**Limitations.** Our method faces challenges when dealing with fine-grained control signals that open-source VLMs struggle to distinguish, such as subtle spatial directions (*e.g.*, moving left versus right). The quality of controllability is also inherently tied to the reasoning and grounding capacity of the underlying VLM, and discrepancies across models directly affect performance. These limitations could be alleviated by targeted finetuning existing VLMs or using stronger VLMs in the future. Moreover, our approach requires access to VLM weights for gradient backpropagation, which, while efficient, prevents its use with proprietary frontier models that do not expose parameters. In this work, we focus exclusively on video diffusion models, leaving other architectures and long-video generation unexplored, which would be interesting future work.

**Future Work.** Beyond test-time training, scaling our VQA paradigm to post-training settings could systematically enhance the prompt-following abilities of video generators. Extending our approach to other video generation architectures, as well as to modalities such as 3D generation and audio-visual synthesis, would further broaden its applicability. As VLMs continue to advance, their improved reasoning ability will naturally strengthen controllability in video generation under our framework, paving the way toward a more general paradigm in which foundation models provide universal supervision across modalities and tasks.

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
