# OpenReview forum: "Controlling Video Generation with Vision Language Models"
_ICLR.cc/2026/Conference — Submitted to ICLR 2026_

### Official Review · Reviewer_SA7h · 2025-10-30

**Soundness:** 3
**Presentation:** 1
**Contribution:** 2
**Rating:** 2
**Confidence:** 5

**Summary:**

This manuscript introduces Ask-A-Video, a novel test-time training paradigm that enhances the controllability of off-the-shelf video generation models without requiring any video training data. The core idea is to frame controllable video generation as a Visual Question Answering (VQA) task. A frozen Vision-Language Model (VLM) acts as a differentiable judge: it analyzes frames generated by the video model, answers control-related questions (e.g., "Is the car red?"), and the resulting VQA loss is directly backpropagated through the VAE decoder to fine-tune the video generator. This approach enables fine-grained spatial, motion, and physical plausibility control through simple text or visual prompts.

**Strengths:**

1. The formulation of controllable video generation as a differentiable VQA task is highly innovative. It provides a unified and flexible control interface that bypasses the need for expensive, modality-specific video datasets and dedicated control modules.
2. The method's ability to work with any off-the-shelf video generator without video data is a significant practical advantage. It lowers the barrier to achieving fine-grained control, making it accessible for users who lack large-scale, annotated video datasets for fine-tuning.

**Weaknesses:**

1. The quality of control is entirely dependent on the VLM's perceptual and reasoning capabilities. This creates a ceiling for the method's performance and means its weaknesses are directly inherited from the frozen VLM, which the method itself cannot improve.
2. The method inherently requires multiple forward/backward passes through both the video generator and a large VLM (7B parameters in the experiments) during test-time training. This introduces significant computational overhead compared to a single inference pass, which may limit its use in real-time or resource-constrained applications. A more detailed analysis of the runtime and memory cost compared to standard inference would be beneficial.
3. The quantitative results are presented as a single table with percentage improvements, but the metric used (e.g., success rate judged by humans? the VLM itself?) is not explicitly defined. Furthermore, comparisons are primarily made against "prompt extension," which is a relatively weak baseline. A comparison against state-of-the-art methods that use dedicated control modules (even if they require video data) would better situate the performance of Ask-A-Video.
4. The paper would be strengthened by ablations that quantify the contribution of key components. For example: How critical is the high-order UniPC solver? What is the impact of the number of frames sampled for VLM input? How does the performance scale with the rank of the LoRA adapters? An ablation on the choice of VLM (e.g., a smaller vs. larger model) would also shed light on the dependency mentioned in Weakness #1.
5. The model's reliance on a single VLM query for supervision may be insufficient for tasks requiring the understanding of temporal dependencies, such as logical sequences (e.g., following a recipe), causal relationships (e.g., a light turning on after a switch is flipped), or basic physical prerequisites (e.g., a door opening before someone walks through it). The current architecture lacks an explicit mechanism to decompose and verify these multi-step constraints.

**Questions:**

1. The VLM is used to judge individual or a small subset of frames. How does the method ensure that the imposed control (e.g., an object's position or a physical law) is enforced consistently across the entire temporal sequence, and not just on the sampled frames?
2. The failure case shows the model "hacking" the VLM by changing the color of feet instead of the position. Could a multi-question or chain-of-thought prompting strategy for the VLM (e.g., "Is there a dot on the ground? Is the person's foot on the dot?") help mitigate such adversarial failures?
3. The method requires access to the VLM's weights for gradient backpropagation. Have you explored any proxy methods or approximations that could make this approach work with proprietary, black-box VLMs (e.g., GPT-4V)?
4. Could you provide more details on the human evaluation process used to generate the quantitative results in Table 1? For instance, how many evaluators were involved, and what were the specific instructions given to them?
5. The paper focuses on short video clips (25-81 frames). How would the memory-efficient strategy scale to generating significantly longer videos, and what new challenges might arise in maintaining coherence and control over a longer horizon?

---

### Official Review · Reviewer_BbDs · 2025-10-31

**Soundness:** 3
**Presentation:** 2
**Contribution:** 3
**Rating:** 4
**Confidence:** 3

**Summary:**

This paper proposes a strategy that enables prompt adherence and physically plausible controllable video generation by using a VLM as a judge. Specifically, instead of existing reward model-based RL approaches, it directly backpropagates VQA loss through DiT + VAE decoder + VLM to update the LoRA of the DiT, where several strategies are proposed to reduce excessive memory usage.

**Strengths:**

- The approach of updating DiT LoRA by directly integrating VLM loss is intuitive.
- The qualitative results presented in the paper are impressive.
- The strategies for saving memory during the process of updating DiT through VAE and VLM with VLM loss are well-designed and efficient.

**Weaknesses:**

- Comparison with existing (e.g., RL-based) works that integrate VLM judges is absent (for example, works discussed in the 'VLM-as-a-Judge' section of Related Work).
- Quantitative evaluation is insufficient. Only a user study is included, and other quantitative metrics such as VBench are not reported.
- Video results are not presented, which makes the qualitative results (frames) in the paper less convincing.

**Questions:**

It would be helpful if VQA loss graph visualization can be provided.

---

### Official Review · Reviewer_5Bmf · 2025-11-01

**Soundness:** 3
**Presentation:** 3
**Contribution:** 3
**Rating:** 6
**Confidence:** 4

**Summary:**

This paper explores an interesting and timely topic, using vision-language models (VLMs) as powerful evaluators for diverse controllable video generation tasks without relying on labeled training data. The proposed method is a test-time fine-tuning approach that formulates video generation as an optimization process by backpropagating the VQA loss from the VLM to the video generator. It requires no additional control modules, supports any input modality, and can be applied to any video generation model without training on video data. Experiments are conducted on both text-to-video (T2V) and image-to-video (I2V) settings across multiple generators, demonstrating the generalizability of the proposed framework. The results show consistent improvements in prompt adherence and visual prompting quality. Furthermore, the learned weights can be distilled into the generator’s parameters for efficient deployment.

**Strengths:**

1. Using VLMs as judges for controllable video generation is an interesting and underexplored direction. The idea of employing VLMs as differentiable supervision signals to guide video generators is novel and promising.
2. The proposed framework is flexible — it can be applied to control any modality that the VLM supports. It does not rely on explicit video or control labels and is compatible with any video generator.
3. The improvements in prompt adherence and physical plausibility shown in Figures 3 and 4 are impressive. The visual prompting results in Figure 5 are also intriguing.
4. The failure cases are informative, illustrating two key weaknesses: (1) the strong visual priors of the generator make it difficult to produce meaningful variations, and (2) the generator may sometimes “trick” the VLM into predicting incorrect but seemingly plausible answers.

**Weaknesses:**

1. As discussed in the related works section, Luo et al. (2025) explored a similar idea of leveraging VLMs for image generation. The paper should better clarify the differences in challenges and methodologies, particularly explaining why directly applying Luo et al. (2025) to the video domain would be problematic.
2. Since the proposed approach is based on a fine-tuning framework, a discussion on computational cost and training time when using different video backbones (e.g., Wan, FramePack, and HunyuanVideo) is needed. This would help readers understand the trade-offs in efficiency and scalability.
3. There are no supplementary videos provided, making it difficult to assess the visual quality of the model.

**Questions:**

1. The paper mentions that four frames are uniformly sampled from each generated video and paired with a VQA prompt. With only four frames, how does the VLM ensure temporal consistency or capture motion-related aspects of the video?
2. In Figure 5, the visual prompting results show a generated hand with a reddish color, which may reveal a limitation of the visual prompting approach. Is there any explanation or potential mitigation for this issue?

---

### Official Review · Reviewer_ZZRB · 2025-11-01

**Soundness:** 3
**Presentation:** 3
**Contribution:** 3
**Rating:** 6
**Confidence:** 3

**Summary:**

The paper presents Ask-A-Video, where a pre-trained VLM is used to improve video generation models by formulating the problem into a VQA problem. The framework freezes the VLM and inputs the prompt and the generated video into the VLM to verify whether the video is generated accordingly to the prompt, and uses this answer to backpropagate the VQA loss to directly optimize the LoRA layers within the video model. The method shows drastic improvements in terms of prompt adherence, and shows that the optimized LoRA weights can be transferred across similar scenes.

**Strengths:**

1. The motivation and the method is very convincing and straightforward. It is also interesting that LoRA layers in video diffusion models can be directly optimized with the VQA loss, and reflects the prompt in the generated video afterwards.

2. The LoRA weights are also transferrable across similar scenes, which mitigates its limitations of requiring optimization for each video.

3. The generated videos with the method shows clear improvements visually, along with the quantitative results. Furthermore, the authors also include discussions for failure cases, where the method fails when prompted to make drastic changes that contradicts its generative prior, which sounds reasonable.

4. The authors also share details about the implementations and efforts for mitigating the computation. While this may seem minor, this adds quite a bit to the paper as practical problems, such as handling outputs from noisy latents or handling memory constraints are commonly encountered when handling video diffusion models and VLMs.

**Weaknesses:**

1. The results does not seem to show the quantitative results in terms of the visual quality for the generated videos. Considering that the method involves training signal from an external VLM, it would be crucial to show that the VQA loss is able to improve prompt adherence without much damaging the visual quality.

2. The method is missing specifications for the time required to optimize the LoRA weights rather than stating "sometimes minutes", where  a more concrete number would help to understand how much the method costs. Similarly, some details such as training steps required for the LoRA weights seems missing.

**Questions:**

1. Would it be possible to group videos with similar actions or objects and train object-wise or action-wise LoRA layers with the method? Given the results from transferring LoRA weights, it seems possible to learn a set of LoRA weights, which can be maybe ensembled in a MoE-fashion to improve video models in general in the future.

---

### Meta-Review · Area_Chair_eN5a · 2026-01-05

**Summary:**

This paper introduced a new controllable video generation methods by formulating it as a VQA problem. It uses a frozen vision language model to answer control-related questions and use update the LoRA weights using the VQA loss. Compared with conventional controllable video generation methods, this method does not require training video data. Reviewers acknowledge that the proposed method is: 1. The proposed method is novel and convincing; 2. The qualitative results show clear improvement; 3. No training video is required is a significant advantage. The major concerns from the reviewers that prevent this paper to be a qualifiable publication is: 1. Lack of quantitative results of the visual quality of the generated videos; 2. The computational cost is missing; 3. Comparisons with RL-based method which uses VLM as judge is missing. 4. No video results are presented. 5. Lack of ablation studies for the design choice of the method. The above concerns from the reviewers are valid and as there is no rebuttal from the authors to clarify the concerns, I tend to reject this paper.

**Reviewer Concerns:**

The authors did not post the rebuttal to address the concerns and questions from the reviewers.

**Reviewer Scores:**

This paper gets score of 6, 6, 4, 2. As the authors did not post rebuttal to address the reviewers' concerns and answer reviewers' questions, the score will remain. As the reviewers's concerns are valid and cannot be addressed, I tend to reject this paper.

---

### Decision · Program_Chairs · 2026-01-26

Reject